# Real-World Evaluation of a Population Germline Genetic Screening Initiative for Family Medicine Patients

**DOI:** 10.3390/jpm12081297

**Published:** 2022-08-08

**Authors:** Megan Leigh Hutchcraft, Shulin Zhang, Nan Lin, Ginny Lee Gottschalk, James W. Keck, Elizabeth A. Belcher, Catherine Sears, Chi Wang, Kun Liu, Lauren E. Dietz, Justine C. Pickarski, Sainan Wei, Roberto Cardarelli, Robert S. DiPaola, Jill M. Kolesar

**Affiliations:** 1Division of Gynecologic Oncology, Department of Obstetrics and Gynecology, University of Kentucky Markey Cancer Center, Lexington, KY 40536, USA; 2Department of Pathology and Laboratory Medicine, University of Kentucky Chandler Medical Center, Lexington, KY 40536, USA; 3Department of Pharmacy Practice and Science, University of Kentucky College of Pharmacy, Lexington, KY 40506, USA; 4Department of Family and Community Medicine, University of Kentucky Chandler Medical Center, Lexington, KY 40536, USA; 5Department of Clinical Research, University of Kentucky Markey Cancer Center, Lexington, KY 40536, USA; 6Shared Resource Facility, University of Kentucky Markey Cancer Center, Lexington, KY 40536, USA; 7Division of Cancer Biostatistics, Department of Internal Medicine, University of Kentucky College of Medicine, Lexington, KY 40506, USA; 8Dr. Bing Zhang Department of Statistics, University of Kentucky, Lexington, KY 40536, USA; 9University of Kentucky Markey Cancer Center, Lexington, KY 40536, USA

**Keywords:** genetic screening, family medicine, pharmacogenomics, real-world

## Abstract

Hereditary factors contribute to disease development and drug pharmacokinetics. The risk of hereditary disease development can be attenuated or eliminated by early screening or risk reducing interventions. The purpose of this study was to assess the clinical utility of germline medical exome sequencing in patients recruited from a family medicine clinic and compare the mutation frequency of hereditary predisposition genes to established general population frequencies. At the University of Kentucky, 205 family medicine patients underwent sequencing in a Clinical Laboratory Improvement Amendments of 1988-compliant laboratory to identify clinically actionable genomic findings. The study identified pathogenic or likely pathogenic genetic variants—classified according to the American College of Medical Genetics and Genomics variant classification guidelines—and actionable pharmacogenomic variants, as defined by the Clinical Pharmacogenetics Implementation Consortium. Test results for patients with pharmacogenomic variants and pathogenic or likely pathogenic variants were returned to the participant and enrolling physician. Hereditary disease predisposition gene mutations in *APOB*, *BRCA2*, *MUTYH*, *CACNA1S*, *DSC2*, *KCNQ1*, *LDLR*, *SCN5A*, or *SDHB* were identified in 6.3% (13/205) of the patients. Nine of 13 (69.2%) underwent subsequent clinical interventions. Pharmacogenomic variants were identified in 76.1% (156/205) of patients and included 4.9% (10/205) who were prescribed a medication that had pharmacogenomic implications. Family physicians changed medications for 1.5% (3/205) of patients to prevent toxicity. In this pilot study, we found that with systemic support, germline genetic screening initiatives were feasible and clinically beneficial in a primary care setting.

## 1. Introduction

Genetic factors are important contributors to disease development. Currently, more than 5000 genes are associated with various genetic disorders. The risk of hereditary disease development can be attenuated or eliminated by early screening or risk-reducing interventions, including surgery, lifestyle modifications, and pharmacotherapies [1,2]. Universal early identification of newborns who inherit an actionable childhood disease is already standard care in the United States of America (USA) and has been a successful public health initiative [3]. For patients undergoing exome or genome sequencing, the American College of Medical Genetics and Genomics (ACMG) recommends that pathologists report incidentally discovered pathogenic or likely pathogenic variants in genes with highly penetrant disease phenotypes. These genes, referred to as the ACMG secondary findings (SF) genes, are primarily associated with cancer and cardiovascular disease and have associated treatment or prevention strategies [4,5,6]. The number of actionable genes has recently increased from 59 (ACMG SF v2.0) [4,5], to 73 (ACMG SF v3.0) [6].

Similarly, germline pharmacogenomic polymorphisms are common and influence the drug pharmacokinetics of absorption, distribution, metabolism, and excretion. Though early identification of pharmacogenomic polymorphisms may prevent severe adverse reactions to common medications and avoid therapeutic failure [7,8], pharmacogenomic screening is not routine in the general population [9]. Currently, the USA Food and Drug Administration (FDA) has identified over 450 drugs that have pharmacogenomic considerations [10]. The Clinical Pharmacogenetics Implementation Consortium (CPIC) helps clinicians and pharmacists navigate this complex genetic information and highlights the level of evidence supporting each pharmacogenomic variant’s importance [11,12]. CPIC provides evidence-based, variant-specific prescribing guidance. Pharmacogenes with sufficient evidence to modify prescribing actions are classified as Level A. Currently, there are 79 different gene–drug interactions comprising 61 drugs and 21 pharmacogenes classified as Level A [11,12].

Advances in sequencing technologies have enabled a population-level approach to genomic medicine. Population-based studies have demonstrated a carrier rate of clinically actionable hereditary disease predisposition variants around 2–4% [13,14]. Many of these patients did not have a genetic diagnosis until participation in these programs [15]. Furthermore, several studies have demonstrated that pharmacogenomic screening initiatives improve patient safety outcomes [8], decrease healthcare costs [16], and decrease the incidence of polypharmacy [17]. While prior studies have evaluated either disease predisposition genes or pharmacogenes, advancements in sequencing allow for the simultaneous assessment of both and can provide a more complete assessment of an individual’s risk of disease, drug toxicity, and optimal medication dosing strategies.

The purpose of this study was to assess the clinical utility of screening family medicine patients for clinically actionable hereditary predisposition and pharmacogenomic germline variants using medical exome sequencing.

## 2. Materials and Methods

### 2.1. Study Design and Setting

This was a single institution prospective cohort study. Patients who received primary care from University of Kentucky family medicine physicians were prospectively enrolled. Participants underwent pre-test education led by a certified genetic counselor to ensure participants understood the test scope and limitations and the lifetime and familial implications of the test results. Patients then underwent germline medical exome sequencing. Sequencing results were reviewed by our multidisciplinary precision medicine team and delivered to each participant—by mail, secure online patient portal or both—and returned to the enrolling physician with recommendations for further follow-up for hereditary disease predisposition and potential impact of pharmacogenetic variants on drug absorption, distribution, metabolism, or excretion. The follow-up of clinical management was obtained from the electronic health record.

### 2.2. Patient Selection

Between November 2018 and September 2020, a convenience sample of patients over 40 years of age with at least one chronic condition managed by their family physician with pharmacotherapy was identified and invited to enroll in this prospective cohort study during routine clinic visits. Patients with known hereditary syndromes were excluded. The family physician informed the patient of the study, and coordinators assisted with enrollment and obtaining written informed consent. This study was conducted pursuant to the guidelines of the Declaration of Helsinki and in accordance with the US Common Rule and was approved by the University of Kentucky Institutional Review Board (IRB protocol #47486 on 7 November 2018).

All participants received germline medical exome sequencing at no cost and a blood sample was obtained for the sequencing. Demographic and clinical data were abstracted from the electronic health record. De-identified data were collected and managed using Research Electronic Data Capture (REDCap) hosted at the University of Kentucky [18,19]. We followed the Strengthening the Reporting of Observational Studies in Epidemiology (STROBE) reporting guideline [20].

### 2.3. Germline Sequencing, Pathogenic Variant Interpretation and Reporting of Results

Germline next generation sequencing using the Agilent Technologies (Santa Clara, CA, USA) medical exome kit was performed in an in-house Clinical Laboratory Improvement Amendments (CLIA) of 1988-compliant laboratory using patient blood samples obtained by phlebotomy. Germline pathogenic and likely pathogenic variants of any of the 59 ACMG SF v2.0 genes and 14 of the 21 genes with CPIC Level A drug recommendations were sequenced and clinically annotated. *CYP2B6* was not included as this was added to the Level A gene–drug list after the study’s commencement. *HLA-**A*, *NUDT15*, and *UGT1A1* variants were not included due to suboptimal probe coverage. Because of the rare clinical usage of peginterferon alfa-2a and 2b, the *IFNL3* and *IFNL4* variants were also not included. Variants for each pharmacogene included in the testing are reported in Table A1. The ACMG SF v2.0 genes and their associated phenotypes are listed in Table A2 [4].

Phlebotomy was used to obtain whole blood samples for germline testing. FASTQ files generated from an Illumina (San Diego, CA, USA) HiSeq 2500 System were aligned to the reference sequence human genome (GRCh37) using a Burrows-Wheeler Aligner (BWA 0.7.8) [21]. Aligned reads were converted to binary alignment map format using Sequence Alignment/MapTools software (V1.8) [22]. Variant calling was carried out using Genome Analysis Toolkit (V4.0.12.0) [23] and VarScan (v2.3.9) [24]. Variants were annotated using Ensembl Variant Effect Predictor (VEP_89) [25] and public databases, including ClinVar [26], 1000 Genomes [27], and the Genome Aggregation Database (gnomAD) [28]. Mutations were reported according to Human Genome Variation Society nomenclature guidelines [29].

### 2.4. Patient Follow-Up and Clinical Impact

Genomic results for hereditary disease predisposition genes and pharmacogenes were reported by the clinical laboratory to the precision medicine team and discussed at biweekly case conferences. A copy of the report was delivered to the enrolling physician, uploaded into the electronic health record, and delivered to each participant via mail, secure online patient portal, or both. The precision medicine team was composed of a multidisciplinary group of clinicians (including the family physicians enrolling subjects), scientists, clinical pharmacists, and genetic counselors. This team made recommendations for follow-up genetic counseling for those with identified hereditary disease predisposition mutations. This team also discussed potential genotype-related therapy modifications for patients with variant pharmacogenes; however, individual patient factors were used to determine need for drug modification. Precision medicine team recommendations were delivered to the enrolling physician via a written recommendation letter. Referrals and therapy modifications were ordered by the enrolling physician. The date of referral and completion of genetic counseling appointment was obtained by review of the electronic health record. Therapeutic decisions resulting from identification of a mutation were also obtained by review of the electronic health record.

### 2.5. Determination of Relevance to Therapy or Disease

The CPIC Level A gene–drug pairs are reported in Table A3 [30]. Clinical relevance of pharmacogenetic polymorphisms to a given patient was determined by electronic health record review. Patient medical problems and medication lists were reviewed to identify therapies relevant to patient pharmacogenomic variants. Physicians reported any medication-related adverse events and any medication changes resulting from study findings to research personnel. A review of the medical record was performed to identify any additional genotype-related toxicities.

### 2.6. Statistical Analysis

Participant demographic and clinical characteristics were assessed using descriptive statistics. Carrier and allelic frequencies obtained from our study population were reported alongside expected population frequencies. Expected American population pharmacogenomic allele frequencies were reported by allele; when American population frequencies were unavailable, expected European population frequencies were used as the Kentucky population is 84% non-Hispanic White [31]. Population pharmacogenomic frequencies were obtained from CPIC [11], Exome Aggregation Consortium (ExAC) database [32], gnomAD [28], or The Trans-Omics for Precision Medicine (TOPMed) Program [33].

Expected population hereditary predisposition gene mutation frequencies were reported by gene and variant. Because many of the variants are rare and data regarding the frequencies in specific populations are often incomplete, global population allelic frequencies were reported. Expected frequencies were obtained from gnomAD_Exome [28], CPIC [11], and TOPMed [33].

Observed hereditary disease predisposition gene mutation carrier frequency was compared to the frequency observed by the Geisinger group’s DiscovEHR study [14]. This study was chosen as a comparison due to its publicly available data, USA population, similar use of the ACMG SF genes, and similar clinical curation and application of clinical laboratory standards suitable for clinical application in a subset of 1415 patients. Though this group used ACMG SF v1.0, which included 56 genes [5], and the present study used ACMG SF v2.0, which included 59 genes [4], no carriers in any of the four non-overlapping genes (*MYLK* removed; *ATP7B*, *BMPR1A*, *SMAD4*, and *OTC* added) were identified. As the Geisinger study only reported carrier frequencies for autosomal dominant conditions or homozygous carriers for autosomal recessive conditions, *MUTYH* heterozygotes were excluded from this comparison.

The observed mutation frequencies were compared to global population allelic frequencies and the observed carrier frequencies were compared to the Geisinger group’s DiscovEHR study [14] using Fisher’s exact test. The *p*-value, odds ratio (OR) and 95% confidence interval (CI) were obtained by using the “fisher.test” function in R. The Hommel method was used for multiple comparison adjustment. An adjusted *p*-value < 0.05 was considered statistically significant. All statistical analyses were performed with R (version 4.1.2).

## 3. Results

### 3.1. Study Population

Of the 215 family medicine patients enrolled in this study, one was later determined ineligible and nine did not provide a blood sample; therefore, 205 patients were eligible for analysis (Figure 1). Demographic characteristics are summarized in Table 1. The median age was 61 (range 51–68) and most (79.5%; 163/205) were non-Hispanic White. Men and women were approximately evenly represented.

### 3.2. Genomic Variant Frequencies

Pharmacogenomic variants were common and are reported in Table 2; 76.1% of the population (156/205) carried at least one pharmacogenomic variant (range per patient: 0–4). *CYP4F2*, *SLCO1B1* and *VKORC1* variants were most common and carried by 52.7 (108/205), 24.9 (51/205) and 14.1% (29/205) of patients, respectively. As shown in Table 3, allele frequencies in our population were similar to those reported in American or European populations; however, the *VKORC1* *-1639G>A polymorphism appeared less frequently (0.0780) in our population compared to the expected American population (0.4643). No participants carried *CACNA1S*, *CFTR*, *CYP2C19*, *HLA-B*, or *RYR1* variant alleles.

Hereditary disease predisposition variants were present in 6.3% (13/205) of the population and are reported in Table A4. Our population had similar autosomal dominant carrier frequencies compared to the Geisinger Group [14] (4.9% 10/205 vs. 3.3% 46/1415; OR 1.53, 95% CI 0.68–3.13, *p* = 0.222). As the Geisinger study only reported carrier frequencies for autosomal dominant conditions or homozygous carriers for autosomal recessive conditions, *MUTYH* heterozygotes were excluded from any comparison to this study. *MUTYH*, *CACNA1S*, and *KCNQ1* gene mutations were most frequent in our population and affected 1.5 (3/205), 1 (2/105) and 1% (2/105) of University of Kentucky patients, respectively. When comparing our population to the Geisinger population at the gene level (e.g., any pathogenic or likely pathogenic mutation in a specific gene) after correction for multiple comparisons, overall carrier frequency was similar among the groups.

Table 4 demonstrates variant frequencies of individual mutations compared to global population frequencies. In the present study, *KCNQ1* variants were identified in higher frequencies than in the gnomAD_Exome study [28], including c.1075C>T (p.Gln359*) (OR 647.16; 95% CI 8.22, 4.50 × 10^15^, *p* = 0.019) and c.1394-1G>T OR 614.648, OR 7.81, 4.50 × 10^15^, *p* = 0.019). Similarly, the *SDHB* variant c.418G>T (p.Val140Phe) was identified in a higher frequency than in the gnomAD_Exome study [28] (OR 204.90; 95% CI 3.89, 2696.96). The present study also identified three rare variants in our population not reported in large population databases—*APOB* c.2477_2478dupTT (p.Leu827Phefs*37), *BRCA2* c.2517C>A (p.Tyr839*), and *DSC2* c.2184dupT (p.Pro729Serfs*2—and two novel *CACNA1S* variants c.4161delC (p.Thr1388Profs*36) and c.930delC (p.Trp311Glyfs*23). The frequency of *MUTYH* variants c.1187G>A (p.Gly396Asp) and c.536A>G (p.Tyr179Cys), *LDLR* c.858C>A (p.Ser286Arg), and *SCN5A* c.4877G>A (p.Arg1626His) were similar to frequencies reported in the gnomAD_Exome study [28] and TOPMed [33].

### 3.3. Clinical Impact: Pharmacogenes

Clinical impact of identification of variant pharmacogenes is detailed in Table 5. Only 4.9% (10/205) patients were prescribed a medication metabolized by a relevant CPIC Level A pharmacogenomic variant. These included four individuals with decreased *SLCO1B1* function receiving simvastatin, one *CYP4F2* *1/*3 patient prescribed warfarin (5 mg/day), four *CYP2C9* intermediate metabolizers prescribed a non-steroidal anti-inflammatory drug (NSAID), and one *CYP2D6* intermediate metabolizer treated with amitriptyline (10 mg/day). The international normalized ratio (INR) remained therapeutic for the patient treated with warfarin and the precision medicine team recommended no genotype-directed drug modifications. The patient treated with amitriptyline also remained stable on the current dose and the precision medicine team recommended against any genotype-directed drug modifications. Family physicians made three medication changes based on pharmacogenetics, all among patients receiving statins. Two patients remained on simvastatin (40 and 20 mg/day) despite identification of decreased *SLCO1B1* function. With the exception of two *CYP2C9* intermediate metabolizers prescribed 2400 mg ibuprofen/day and who experienced concurrent gastrointestinal symptoms requiring treatment with a proton pump inhibitor (omeprazole and pantoprazole), no associated pharmacologic adverse effects were identified in these 10 patients.

### 3.4. Clinical Impact: Hereditary Predisposition Genes

Most patients, 69.2% (9/13), with a hereditary gene mutation were referred to genetic counseling as recommended by the precision medicine team (Figure 2). Referrals were sometimes problematic due to genetic counseling staff turnover and a lack of expertise for very rare mutations. As of February 2022, five of nine patients (69.2%) had attended this appointment and all patients completing genetic counseling underwent additional clinical interventions. One patient with a likely pathogenic *DSC2* mutation is awaiting a genetic counseling appointment but completed clinical interventions after a cardiology referral from his family physician.

Clinical interventions after identification of germline mutations included familial cascade testing for patients with *BRCA2* (*n* = 1) and *SCN5A* (*n* = 1) germline mutations, cardiac workup and lifestyle modifications for patients with *SCN5A* (*n* = 1), *KCNQ1* (*n* = 1), and *DSC2* (*n* = 1) germline mutations, initiation of early colorectal cancer screening for a *MUTYH* carrier based on carrier status and family history (*n* = 1), and more complete hereditary cancer syndrome germline testing due to a suggestive family history for an *MUTYH* carrier (*n* = 1).

## 4. Discussion

Our results demonstrated the feasibility, and suggested a potential benefit, of a population-level genomic screening program for patients cared for in a family medicine clinic. To our knowledge, this is the first to combine hereditary predisposition testing with pharmacogene assessment in one institution with comparisons to general population frequencies. Similar to the Geisinger group [14] and the United Kingdom (UK) Biobank [13], we evaluated individuals for the presence of at least one mutation in all of the ACMG SF v2.0 genes [4,5,6]. We identified a similar frequency of carriers of medically actionable mutations in our cohort to compared to the Geisinger group [14]. Although we did not directly compare our findings to the UK Biobank [13], our mutation frequency exceeded the UK Biobank study’s observed mutation frequency (2.0%).

Kentucky is reported to have the highest incidence of, and mortality from, cancer in the USA [34], a lower-than-average life expectancy [35], and a higher-than-average cardiovascular disease mortality [36], highlighting the necessity of implementation of a genetic screening program in our state. Early identification of genetic diseases may improve disease outcomes, specifically for cancer and cardiovascular disease prevention.

Similar to the benefit carriers identified by the Geisinger group [15] experienced, the majority of identified carriers in the current study also underwent additional clinical interventions; however, the rate of referral for genetic counseling and additional follow-up practices was lower than expected. At our center, patients may be referred to a cancer-focused genetic counselor, a non-cancer focused genetic counselor, or a cardiologist depending on the hereditary mutation identified. Outside of cancer, referrals were sometimes problematic due to staff turnover and a lack of expertise for certain very rare mutations. Assuring adequate expertise, staff, and a clear referral process are important considerations for implementing a genetic screening program and could have improved the referral rate in this study. Additionally, several patients at risk for hereditary diseases who were referred for genetic counseling did not attend the appointment. At the time of this writing, one patient remains awaiting an appointment. We speculate this could be related to the increasing clinical demand of genetic counselors [37] and wait times for an appointment [38]. Prolonged wait times have been associated with no-show rates to genetic counseling appointments [39], and this phenomenon may have contributed to the lower-than-expected completion rates of genetic counseling as some of the study time occurred during the coronavirus disease 2019, which affected appointment wait times, including those for genetic counseling services [40].

Because risks of adverse drug reactions may be attenuated with genotype-directed dosing strategies, pharmacogenomic considerations were added to the USA FDA drug labeling for certain drugs in 2009 [41]. Several studies have demonstrated pre-emptive pharmacogenomic testing initiatives benefit diverse patient populations [8,42,43]. In the present study, at least one pharmacogenomic variant in most patients was identified; however, relevant medication use was infrequent among individuals with an actionable genotype. Because the enrolling physicians used alternative strategies to mitigate adverse drug effects—prescribing simvastatin doses lower than 80 mg to avoid myopathy [44], warfarin titration based on INR [45], and concurrent prescription of proton pump inhibitors for patients treated with NSAIDs [46]––only three patients received genotype-directed medication adjustments. Several patients with variant genotypes were not recommended by the precision medicine team to undergo genotype-directed drug adjustments in the event they were stable on a long-term prescription. While the impact of genotyping was small over the short duration of this study, additional benefits may accrue over time as individuals may start new medications with pharmacogenetic implications and dose adjustments are made prior to their initiation.

Strengths of this research include its prospective nature and real-world implementation. In addition, hereditary disease predisposition genes and pharmacogenes were evaluated and reported concurrent to direct clinical management as a single test and delivered in a single report. Testing was performed in a CLIA-compliant laboratory of a regional academic medical center and results were integrated into clinical practice, which suggests this approach could be feasible in a community setting. Additional strengths of this genomic testing initiative include its integration with a pre-test educational intervention and return of results directly to each patient, which provides an opportunity for patients to consider their genotype when starting new medications. This was a single institution study with a small sample size, which limited the precision of allelic frequency estimates. As clinical pharmacogenomic testing varies, we acknowledge there are many resources that provide guidance for testing and medication dosing strategies. The list of CPIC Level A pharmacogenes may differ from other resources and is not comprehensive. Future pharmacogenomic testing initiatives may use a more comprehensive list of pharmacogenes derived from multiple resources. Additionally, the bioinformatics pipeline used in this study to classify pharmacogenes was not built to address copy number variants, which could be evaluated in future research. Another limitation is the older age of the population enrolled in this study. Most hereditary syndromes manifest at a younger age and enrolling younger patients may have identified more carriers and had a greater impact. Selection bias may have been present as enrolled patients may have had unusual or unexplained phenotypes.

## 5. Conclusions

This study used a prospective, real-world approach to investigate the landscape of mutations in hereditary disease predisposition genes and pharmacogenomic polymorphisms in the family medicine setting of a regional academic medical center. We also evaluated the impact of this screening results on patient management by assessing clinical decision making for individual patients. Our results demonstrated that such a screening program is feasible and that identification of hereditary predisposition gene mutations could be clinically beneficial. While the benefit of pharmacogenetic testing was minimal over the short time period of this study, integrating pharmacogenomic test results into the electronic health record may benefit these patients when starting new medications.

## Figures and Tables

**Figure 1 jpm-12-01297-f001:**
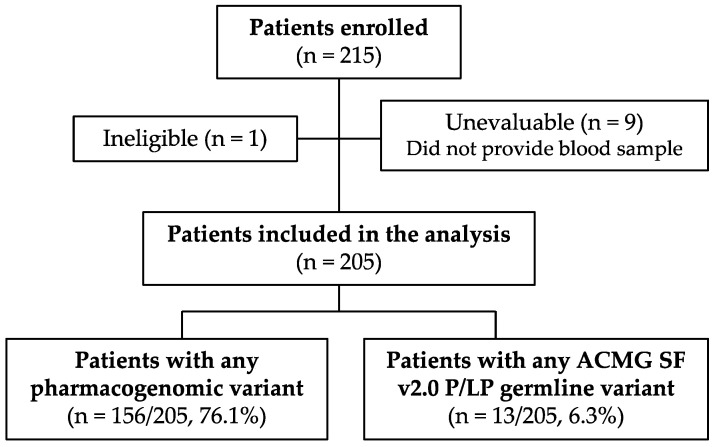
Study flow diagram.

**Figure 2 jpm-12-01297-f002:**
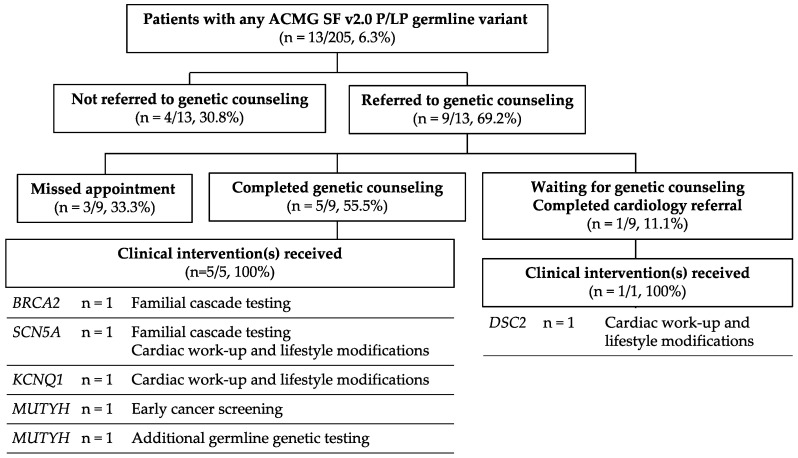
Clinical follow up for hereditary disease predisposition carriers.

**Table 1 jpm-12-01297-t001:** Demographic characteristics of family medicine patients.

Characteristic	Patients n (%)
Total	205
Age (median, IQR)	61 (51–68)
**Race**	
AI/AN	2 (1.0%)
Asian	4 (2.0%)
Non-Hispanic Black	31 (15.1%)
Hispanic White	5 (2.4%)
Non-Hispanic White	163 (79.5%)
**Gender**	
Female	109 (53.2%)
Male	96 (46.8%)

Abbreviations: IQR: Intra-quartile range; AI/AN: American Indian/Alaska Native.

**Table 2 jpm-12-01297-t002:** Pharmacogenomic genotype frequencies of family medicine patients. There were no carriers of *CACNA1S*, *CFTR*, *CYP2C19*, *HLA-B*, or *RYR1* variant alleles.

Pharmacogenomic Variant Gene	Carrier Frequency	Genotype, *n*
Any	156 (76.1%)	
*CYP4F2*	108 (52.7%)	*3/*3: 19 *1/*3: 89
*SLCO1B1*	51 (24.9%)	*15 or *17 ^a^ / *15 or *17: 3 *1/*15 or *17: 42 *1/*5: 6
*VKORC1*	29 (14.1%)	*1173C>T/*1173C>T: 3 *1/*1173C>T: 26
*CYP2C9*	19 (9.3%)	*1/*3: 19
*CYP3A5*	10 (4.9%)	*1/*6: 5 *1/*7: 5
*DPYD*	10 (4.9%)	*1/*c.2846A>T: 2 *1/*2A: 2 *HapB3/*HapB3: 1 *1/*HapB3: 5
*TPMT*	9 (4.4%)	*1/*3A: 6 *1/*3B: 1 *1/*3C: 2
*CYP2D6*	8 (3.9%)	*1/*6: 8
*G6PD*	3 (1.5%)	Class III deficiency: 3

^a^ The differentiating allele of *15 and *17 (NC_000012.12:g.21130388G>A; rs4149015) is not covered by the current NGS enrichment probe; therefore, these variants are reported as *15 or *17. Both *15 and *17 alleles have the same functional status (decreased function) for the *SLCO1B1* gene.

**Table 3 jpm-12-01297-t003:** Pharmacogenomic variants, observed allele frequencies in University of Kentucky family medicine patients, and expected population allele frequencies. There were no variant *CACNA1S*, *CFTR*, *CYP2C19*, *HLA-B*, or *RYR1* alleles in our population.

Pharmacogenetic Variant	Observed Allele Frequency	Expected Allele Frequency ^a^
*CYP4F2*		
*3	0.3098	0.4108
*SLCO1B1*		
*15 or *17 ^b^	0.1171	0.1214 (*15); 0.0519 (*17)
*5	0.0146	0.0224
*VKORC1*		
*-1639G>A	0.0780	0.4643
*CYP2C9*		
*3	0.0463	0.0301
*CYP3A5*		
*6	0.0122	0.0015
*7	0.0122	0.0000
*DPYD*		
*c.2846A>T	0.0048	0.0037
*2A	0.0048	0.0079
*HapB3	0.0171	0.0237
*TPMT*		
*3A	0.0146	0.0343
*3B	0.0024	0.0027
*3C	0.0048	0.0047
*CYP2D6*		
*6	0.0195	0.0025
*G6PD*		
A-202A_376G-III	0.0073	0.0–0.034 ^c^

^a^ When available, American population frequencies are reported; however, because genomics can vary by race and ethnicity and the Kentucky population is 84% non-Hispanic White [31], expected European frequencies may be reported if American frequencies are not available. ^b^ The differentiating allele of *15 and *17 (NC_000012.12:g.21130388G>A; rs4149015) are not covered by the current NGS enrichment probe; therefore, these variants are reported as *15 or *17. Both *15 and *17 alleles have the same functional status (decreased function) for the *SLCO1B1* gene. ^c^ Caucasian prevalence of this *G6PD* variant is 0.0; however, prevalence of any *G6PD* variant in the Americas is 0.034.

**Table 4 jpm-12-01297-t004:** Hereditary predisposition gene mutations, observed allele frequencies in University of Kentucky family medicine patients, and expected global population allele frequencies.

	Allele Frequency				
Gene and Variant	Observed	Expected ^a^	*p*-Value	OR (95% CI)	Adjusted *p*-Value ^b^
*MUTYH* c.1187G>A (p.Gly396Asp) c.536A>G (p.Tyr179Cys)	0.004878 0.002439	0.003027 0.001535	0.353 0.468	1.61 (0.20, 5.90) 1.59 (0.04, 8.96)	0.468 0.468
*CACNA1S* c.4161delC (p.Thr1388Profs*36) c.930delC (p.Trp311Glyfs*23)	0.002439 0.002439	Novel ^c^ Novel ^c^	- -	- -	- -
*KCNQ1* c.1075C>T (p.Gln359*) c.1394-1G>T	0.002439 0.002439	1/264,690 ^d^ 1/251,392	0.003 0.003	647.16 (8.22, 4.50 × 10^15^) 614.65 (7.81, 4.50 × 10^15^)	0.019 0.019
*APOB* c.2477_2478dupTT (p.Leu827Phefs*37)	0.002439	Rare ^e^	-	-	-
*BRCA2* c.2517C>A (p.Tyr839*)	0.002439	Rare ^e^	-	-	-
*DSC2* c.2184dupT (p.Pro729Serfs*2)	0.002439	Rare ^e^	-	-	-
*LDLR* c.858C>A (p.Ser286Arg)	0.002439	0.001077 ^d^	0.358	2.27 (0.06, 12.81)	0.468
*SCN5A* c.4877G>A (p.Arg1626His)	0.002439	10/251,192	0.018	61.41 (1.41, 431.59)	0.071
*SDHB* c.418G>T (p.Val140Phe)	0.002439	3/251,418	0.006	204.90 (3.89, 2696.96)	0.033

^a^ Expected global allele frequencies were obtained from gnomAD_Exome [28] unless otherwise specified. ^b^ Hommel’s multiple testing method was used to adjust for multiple comparisons. ^c^ This is a novel variant and has not been previously reported in large databases but is likely pathogenic. ^d^ Allele frequency obtained from Trans-Omics for Precision Medicine (TOPMed) [33]. ^e^ This is a rare variant and is not reported in large databases but has been previously reported.

**Table 5 jpm-12-01297-t005:** Clinical impact after identification of pharmacogenomic variants.

Variant Pharmacogene Carriers (*n*)	Drug Class	Patients Prescribed Drug in Class (*n*)	Specific Drug Prescribed	Patients Prescribed Specific Drug (*n*)	Toxicity? (*n*)	Drug Change? (*n*)
*CYP2C9* (19)	AC	1	Warfarin	0	-	-
	NSAID	7	Meloxicam Ibuprofen	1 3	0 2 ^a^	0 0
	AED	2	Phenytoin Fosphenytoin	0 0	- -	- -
*CYP2D6* (8)	Narcotic	1	Codeine	0	-	-
	TCA	1	Amitriptyline	1	0	0
	SSRI	2	Paroxetine	0	-	-
*CYP4F2* (108)	AC	9	Warfarin	1	0	0
*SLCO1B1* (51)	Statin	27	Simvastatin	4	0	3 ^b^
*VKORC1* (29)	AC	1	Warfarin	0	-	-

^a^ Two *CYP2C9* intermediate metabolizers experienced gastrointestinal side effects when prescribed ibuprofen and were managed with concurrent proton pump inhibitor therapy. ^b^ Three drug changes were made resulting from identification of decreased *SLCO1B1* function, including simvastatin to atorvastatin, simvastatin to rosuvastatin, and atorvastatin to pravastatin. Abbreviations: AC: anticoagulant; NSAID: non-steroidal anti-inflammatory drug; AED: anti-epileptic drug; TCA: tricyclic antidepressant; SSRI: selective serotonin reuptake inhibitor.

## Data Availability

The authors received no special privileges in accessing the data. Raw data cannot be shared because they are both potentially identifying and contain sensitive patient data, including geographic location and specific dates of testing, clinical interventions, and receipt of a drug.

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
