# Peer review of "Real-World Evaluation of a Population Germline Genetic Screening Initiative for Family Medicine Patients"

_jpm, 2022, doi:10.3390/jpm12081297_

Round 1

Reviewer 1 Report

Real-world evaluation of a population germline genetic screening initiative for family medicine patients by Hutchcraft et al is a well written, thoughtfully designed, and relevant study to the fields of medical genetics and pharmacogenomics. While it is generally thought that preemptive genotyping may be most beneficial in these scenarios, there are a number of barriers to its implementation. The data reported in this study helps to provide evidence of the value of genotyping for disease and pharmacogenomic specific variants and highlights several important considerations on the topic.

The main question I have is whether the patients who were genotyped have access to their genetic test results. If not, why? If yes, were they offered an opportunity to review these results with an expert if they had questions? It may be helpful to add this as a discussion point as well, since exome genotyping of patients, especially at younger ages (as potentially recommended by the authors) brings up a number of ethical concerns, which also need to be addressed and planned for if this type of genotyping is to become broadly used.  In addition, if they move or switch doctors, it would be important for them to be able to provide any future physicians with their genotype results. Since germline variants will not change in the lifetime of the patient, explaining this to them and what their test results are (even if they are not currently taking a medication that is affected by their genotype or if their current prescribing physician did not feel a clinical intervention was necessary) and what it means for current or future medication use is an essential factor.

Reviewer 2 Report

This article shows a pilot study analyzing particular variants related to either pharmacogenetics or disease-related in a population attending family medicine clinic. In addition, the authors describe the actions that were followed up after discovering any mutation.

Although the manuscript reflects a tremendous work, there are a few items that need to be addressed:

1. In the introduction the authors say that PGx is related to drug metabolism, but that is inaccurate, since PGx variants can affect not just metabolism, but also other steps of the ADME, as well as pharmacodynamics (like VKORC1).

2. In the same section, they mention about drug toxicity, but some variants could cause therapeutic failure, and that is not considered either.

3. Looking at methodology /results, it is unclear why the results tables register variants results for 9 genes, while the appendix A lists 14. In addition, is unfortunate the selection of variants/SNPs for some of the genes, like in the case of CYP2C19 (only one), CYP2C9 (not including *2), or worse, CYP2D6 *50, variant which CPIC and PharmGKB has it considered as level 3.

I would suggest the authors to genotype patients for polymorphisms that are considered as level 1 (A), and are guidelines associated with it.

4. When comparing table 2 and with table 5, there is some confusion. While in table 2 the homozygous and heterozygous for a particular variant are separated, in table 5 are combined, and it is hard to interpret which patients had to be adjusted.

5. In the methods it is unclear if a genetic counselor was used when patients were recruited for the study. (It seems to make more sense to explain to patients the potential risks, specially since the study expanded to hereditary disease predisposition, and could affect not only the patient but their families).

6. On section 3.4- "most patients" could authors clarify why "most" and  not "al"l patients with hereditary disease predisposition polymorphisms were not referred to genetic counseling?
